# Alcohol Consumption during a Pandemic Lockdown Period and Change in Alcohol Consumption Related to Worries and Pandemic Measures

**DOI:** 10.3390/ijerph18031220

**Published:** 2021-01-29

**Authors:** Silvia Eiken Alpers, Jens Christoffer Skogen, Silje Mæland, Ståle Pallesen, Åsgeir Kjetland Rabben, Linn-Heidi Lunde, Lars Thore Fadnes

**Affiliations:** 1Department of Addiction Medicine, Haukeland University Hospital, 5021 Bergen, Norway; linn-heidi.lunde@uib.no (L.-H.L.); lars.fadnes@uib.no (L.T.F.); 2Department of Clinical Psychology, Faculty of Psychology, University of Bergen, 5020 Bergen, Norway; 3Department of Health Promotion, Norwegian Institute of Public Health, 5015 Bergen, Norway; jens.christoffer.skogen@fhi.no; 4Department of Public Health, Faculty of Health Sciences, University of Stavanger, 4021 Stavanger, Norway; 5Alcohol & Drug Research Western Norway, Stavanger University Hospital, 4010 Stavanger, Norway; 6Department of Global Public Health and Primary Care, Faculty of Medicine, University of Bergen, 5020 Bergen, Norway; silje.maeland@uib.no; 7Research Unit for General Practice in Bergen, The Norwegian Research Centre, NORCE, 5008 Bergen, Norway; 8Department of Psychosocial Science, Faculty of Psychology, University of Bergen, 5020 Bergen, Norway; staale.pallesen@uib.no; 9Optentia Research Focus Area, North-West University, Vanderbijlpark 1900, South Africa; 10Section for Strategy and Analysis, Bergen Municipality, 5020 Bergen, Norway; asgeir.rabben@bergen.kommune.no

**Keywords:** COVID-19, lockdown, alcohol consumption, risk factors, social distancing, pandemic, SARS-CoV-2

## Abstract

Whether lockdown related to the COVID-19 pandemic influences alcohol consumption is not well known. This study assesses alcohol consumption and hazardous drinking behavior during the initial phase of pandemic measures in Norway and identifies potential risk factors. A cross-sectional study (*N* = 25,708) was conducted in Bergen, Norway, following the first six weeks of strict infection control measures. In a model of self-assessed increased alcohol consumption, logistic regression analysis was conducted with independent variables for COVID-19-related worries, joblessness, quarantine, self-reported drinking behavior, age, gender, and occupational situation. These are reported with odds ratios (ORs) with 95% confidence intervals. Fifty-one percent of respondents reported economic or health-related worries due to COVID-19, 16% had been in quarantine, 49% worked/studied from home, 54% reported hazardous drinking behavior, and 13% reported increased alcohol consumption. People aged 30–39 years had elevated odds of increased alcohol consumption during lockdown (OR 3.1, 2.4−3.8) compared to the oldest adults. Increased drinking was more frequent among people reporting economic worries (OR 1.6, 1.4−1.8), those quarantined (OR 1.2, 1.1−1.4), and those studying or working at home (OR 1.4, 1.3−1.6). More than half of respondents reported hazardous drinking behavior. Increased alcohol consumption during lockdown was common among people with economic worries, people in quarantine, and people studying or working at home. These data could be important when adjusting pandemic measures.

## 1. Introduction

The severe acute respiratory syndrome coronavirus 2 (SARS-CoV-2) causing the coronavirus disease 2019 (COVID-19) represents an ongoing global health crisis and pandemic. The state of emergency that it has created is without parallel in recent times [1]. Worldwide, most countries have taken stern measures to get the situation under control, including lockdowns and quarantine. As communities were closing down, concerns were raised about the impact of these measures on other aspects of public health, including mental health [2,3]. The impact of COVID-19 on psychological symptoms and disorders, substance use, and addiction is currently being investigated. Although the empirical evidence is still minimal, most studies so far suggest poorer psychological well-being in the general population due to the COVID-19 emergency [4]. Population-based studies are attesting to this, revealing a high occurrence of self-reported psychological distress symptoms during the early phases of the pandemic across various populations [5,6,7].

COVID-19 has caused major disruptions in the daily life of most people in affected areas and could also impact drinking habits. Both acute and chronic stress are documented risk factors of increased alcohol use [8,9]. Increased alcohol use can thus be regarded as a response to a crisis as well as a coping mechanism. Infection control measures, like physical or social distancing, can also cause higher alcohol use. In the Bergen municipality, the following pandemic measures were implemented on 12 March 2020: social distancing, travel restrictions, post-travel quarantine, closed social arenas, and canceled recreational and cultural events. Schools and universities were closed. Companies advised or ordered their employees to work from home. People were advised to stay at home and avoid others as much as possible to help prevent the spread of COVID-19. For many, this led to social isolation, which in turn can be hypothesized to lead to higher alcohol use at home due to stress, lack of social contact, and boredom. However, there is insufficient data on the effects of pandemic measures on worries and alcohol consumption.

Large-scale disasters, such as a pandemic, can pose major public health threats and need swift solutions that tackle immediate consequences of emergencies [10]. Collecting relevant information as a foundation for potential solutions is crucial. Thus, conducting epidemiological surveys during disasters is important in decision making and in addressing the effects of emergencies. By gathering data related to the same topic from around the globe we can identify different aspects of the same phenomenon and gain understanding from several perspectives and sources. This study relates to a series of articles discussing alcohol consumption during the COVID-19 pandemic [11,12,13] and helps to increase the level of knowledge on this topic.

Specifically, we aimed to investigate various patterns of alcohol consumption and its association with COVID-19 related impacts and worries through a large-scale population-based study in the Bergen municipality in Norway. Our main objectives were:to assess changes in alcohol consumption and hazardous drinking behavior during the initial phase of measures against the COVID-19 pandemic;to identify potential risk factors, such as worries, quarantine, and joblessness, for self-assessed increases in alcohol consumption.

## 2. Materials and Methods

In April 2020, a random sample of 81,170 out of 224,000 adult residents in Bergen, Norway, were invited to participate in a survey concerning the consequences of the COVID-19 measures. The data collection took place between April 15 and April 30. The sample and their contact information were extracted from the National Population Registry of Norway and the common contract register. An electronic online survey questionnaire was distributed via SurveyXact (provider of online survey services). Up to two reminders were sent via SMS and email to those who did not respond. The questionnaire took 15–20 min to complete. At the time of data collection, several restrictions due to COVID-19 had been initiated. These included social distancing, closure of educational, cultural, and training/sport/gym facilities, requirements to work from home, and introduction of quarantine requirements. No changes in the restrictions took place during data collection. In total, 29,535 (response rate 36%) persons consented to participate in the study.

The questionnaire included items for demographic information, self-reported weight and height, and questions about various aspects of life and health amid the COVID-19 lockdown. The following background variables were included in the analyses of the present article: age, gender, educational attainment, work situation, household income, concerns about infection for themselves and related persons, and concerns about consequences of COVID-19 for their work and economic situation. The main outcome variables in the present study were alcohol consumption and self-assessed change in alcohol consumption.

Alcohol consumption was assessed by the Alcohol Use Disorders Identification Test Consumption (AUDIT-C) which consists of the first three questions of the AUDIT [14,15]. The three questions of AUDIT-C all concern alcohol consumption: frequency of drinking, typical quantity consumed, and frequency of heavy drinking (Appendix B). Each question is scored using a five-point scale (from 0 to 4), thus the composite score of the AUDIT-C ranges from 0 to 12. The AUDIT-C has been shown to possess adequate psychometric properties [16,17].

In the present study, we used an AUDIT cut-off score of 3 for women and 4 for men. AUDIT-C cut-off scores for identifying alcohol consumption above recommended limits vary by population, setting, and culture [18]. A typical cut-off for the determination of hazardous drinking is a score of 3 or greater from a possible 12 using the AUDIT-C [19]. Optimal screening thresholds for alcohol misuse broken down by gender are ≥4 among men and ≥3 among women [20]. Several studies support this assessment: for males, a cut-off score of four appears to be the appropriate choice to identify hazardous drinking. To identify those likely to be diagnosed with an alcohol use disorder, a cut-off score of 5 would be preferable. For females, a cut-off score of 3 seems to be suitable for detecting hazardous drinking; 4 for diagnosable use disorders [16,21]. Generally, the higher the AUDIT-C score, the more likely it is that the person’s drinking is negatively affecting their health and safety and the greater the risk of developing alcohol-related problems, including abuse and dependence [22].

In addition to hazardous drinking, we calculated the consumed mean units per week via units per drinking day and drinking days per month (questions 1 and 2 of the AUDIT-C; Appendix B). Drinking more than six units on one occasion (question 2) at least once a month (question 3) was categorized as frequent binge drinking. Self-reported change in consumption during the lockdown was assessed by a single question comparing current drinking to the time before with three response alternatives (how has your alcohol consumption changed during the period of pandemic measures? Consumption has “decreased”, “not changed” or “increased”). The items dealing with measures against the COVID-19 pandemic (“placed in quarantine”, “temporarily laid-off”, “home office/study”, and “COVID-19 symptoms”) were designed as dichotomous nominal variables and formulated as true/false statements. The questionnaire included two questions on economic worries: one ascertained perceived worry about job loss and the second concerned worry about private financial situations. The responses were recorded on a three-point scale with the response alternatives “strongly agree”, “agree”, and “disagree”. The cut-off (affected) for the variable “economic worries” was set to answering at least one of the two questions with “strongly agree”. The “health worries” variable reflected the items concerning questions on how COVID-19 may affect one’s own or others’ health. Each of these items consisted of a statement to which respondents were to indicate their level of agreement by choosing one of three responses (“strongly agree”, “agree”, and “disagree”). If at least one of the questions was answered with “strongly agree” the variable “health worries” was assigned a score of 1, otherwise the score was set to 0.

Data were analyzed with Stata SE, version 16 (StataCorp, College Station, TX, USA). The response rate was higher among older compared to younger participants, among women compared to men, and among those with high education compared to those with lower education. Therefore, inverse probability weights were used to correct for this (by weighting up underrepresented groups) in the final estimates. The weights were calculated using binomial regression models (generalized linear model (GLM)) and the average weight was 1.0 with a standard deviation of 0.25. Weighted estimates for experiences with various COVID-19 related measures and consequences are presented with their corresponding 95% confidence intervals (CI). Results from descriptive analyses are shown in terms of contingency tables, including medians. Chi-square tests were used to test for statistically significant differences between groups of categorical variables.

A multivariable binary logistic regression model utilizing odds ratios (ORs) and confidence intervals was used to estimate self-assessed increased alcohol consumption. The exposures were COVID-19 related worries and lockdown consequences of pandemic measures adjusted for sociodemographic factors. Significance levels of *p* < 0.05 was used. Not everyone who consented completed the entire questionnaire. Therefore, the number of valid answers varied somewhat across analyses. We based the analyses in this article on questionnaires that had valid responses to all alcohol-related variables (*n* = 25,708; 32% of the total sample). Table 1 provides a descriptive overview of included variables with the number and percentage distributed by age. Descriptive analyses utilizing arithmetic means (including standard deviation) and medians (with 25th and 75th percentiles) were conducted. Gender–age interactions were also tested in the regression models.

The participants provided their consent to participate in the study by accessing the survey (“Yes, I agree to participate in the survey as described on the previous page”). The project was approved by the Regional Committee for Medical and Health Research Ethics, Health Region West (ethics registration code 2020/131560), and was conducted in close dialogue with a data protection official from the University of Bergen. A data protection impact assessment was completed for this project before data collection was initiated.

## 3. Results

### 3.1. Study Population

Among the participants, 56% (14,452/25,708) were women and the median age was 50 years (interquartile range (IQR) 36−63), 40% (10,246/25,573) had more than three years of university or college education, 94% (24,274/25,708) were Norwegian citizens, 87% (19,840/22,920) had an adjusted household income above EUR 25,000 (EUR 1 ≈ NOK 10) per person, 68% (17,447/25,708) were employed/worked, and 8% (2011/25,708) were students. Two-thirds (16,833/24,950) lived with 1−3 other people (Table 1).

### 3.2. Alcohol Consumption

In total, 91% (23,319/25,708) of respondents reported consumption of alcohol. A total of 54% (7804/14,452) of the women reported use of alcohol above the cut-off score for hazardous alcohol use (Table 2, Figure 1). Similarly, for men, 52% (5888/11,256) reported alcohol consumption above the cut-off score. With a higher threshold of 5/6, 16/20% of women/men were considered to show hazardous alcohol use (Appendix A). Regarding mean alcohol consumption, men had a higher overall consumption (4.0 units/week) compared to women (2.4 units/week). In both groups, the consumption was highest in the age groups of 18–29 and 60–69 years. Frequent binge drinking (i.e., drinking more than six units on one occasion at least once a month) was reported by 14% (3684/25,619) of the respondents. The age group of 18−29 years had the greatest percentage of frequent binge drinkers (30%). Men were more likely (up to three times more likely) to be frequent binge drinkers compared to women in all age groups. Self-assessed increased alcohol consumption during the lockdown period was more frequently reported by people reporting economic worries (OR 1.6, 95% CI: 1.4–1.8), in quarantine (OR 1.2, 95% CI: 1.1–1.4), and working or studying from home (OR 1.4, 95% CI: 1.3–1.6, Table 3). Compared to the oldest adults (70+ years) during the COVID-19 lockdown, all other age groups more often reported self-assessed increased drinking; those of 30−39 years of age reported self-assessed increased drinking most often (OR 3.1, 95% CI:2.4–3.8). Decreased drinking during the lockdown was most common among the youngest adults (adjusted OR 4.8, 95% CI: 4.2–5.5, Appendix A), people temporarily laid-off, and those working or studying from home. In general, the youngest participants were most likely while the oldest participants were least likely to change their alcohol consumption (Appendix A). The older age groups were less likely than younger age groups to increase their alcohol consumption while having health-related worries (Figure 1). Moreover, a reduced amount of frequent binge drinking was reported amid respondents with health-related worries in all age categories.

## 4. Discussion

This large population-based study presents novel data on self-assessed changes in alcohol consumption and its associations with pandemic measures and worries during the initial phase of the COVID-19 lockdown in Norway. Worries concerning COVID-19 health-related consequences were associated with slightly increased alcohol consumption. Worries related to economic outcomes had a stronger association with higher alcohol use than health-related worries. Paradoxically, many of the same factors associated with increased alcohol consumption were also associated with reduced alcohol consumption. Thus, we believe that COVID-19-related consequences, such as being temporarily laid-off, home office/study, quarantine and health and economic worries, were associated with a change in alcohol consumption (either up or down), while those not heavily impacted by COVID-19 changed their behavior to a lesser degree.

The unemployment rate in Norway was four times higher during our data collection than immediately before the COVID-19 lockdown [23]. Job loss leads to lower psychological and physical well-being [24]. Thus, being unemployed or temporarily laid-off may result in financial insecurity, which in turn increases economic worries. In relation to our findings of increased alcohol consumption among people laid-off from work, this is worrying.

People in their thirties reported the highest increase in alcohol use. This group is perhaps the most vulnerable in terms of financial dependence. They have generally not been employed for a long period, have many years left before retirement, and often are in more vulnerable economical situations, typically due to mortgage and family size. Other studies have also reported somewhat similar findings suggesting that economic uncertainty combined with unpredictability related to the COVID-19 pandemic can cause anxiety and stress [7], which in turn can lead to increased alcohol consumption [8]. Parallel findings have also previously been observed during economic crises and in response to unemployment and reduction in income; an increase in alcohol consumption was reported, especially among men [25].

Further, alcohol use disorders have been reported to be linked to several types of stress exposure [26], including natural disasters, terrorist events, job loss, divorce, and other personal life stressors [27,28,29]. Our findings are also relatively parallel to those of two studies on changes in alcohol consumption during the COVID-19 pandemic in the United States and Poland, respectively [30,31]. Due to some variations in the degree to which the measures were implemented among different parts of the population we examined, our study increases understanding by specifying associations with various COVID-19-related consequences, such as quarantine and social distancing.

Increased drinking during the lockdown period was also reported by people in quarantine and those with a home office. One can assume that more people started drinking alcoholic beverages at home because of social distancing measures. Drinking alone, rather than in social settings, can result in greater alcohol consumption [32]. On the other hand, physical distancing, closed restaurants and bars, canceled events (culture and sports), and home study were measures which might explain some of the decreased drinking among the youngest adults, as during the lockdown they did not have access to the social settings in which drinking typically takes place, especially for this age group. Private parties and gatherings were also restricted as the number of people one was allowed to have contact with was limited.

In 2018, a public health survey was conducted in the same region with similar inclusion criteria and a similar assessment method (AUDIT-C) as the present study. Then, about 20% of the respondents reported that they had drunk six or more units on the same occasion during the previous 12 months [33]. This corresponds with the results from the current sample, in which 17% reported binge drinking (drinking more than six units on one occasion at least once a month). In both samples, this proportion was about twice as high among men as among women. We also detected the same age gradient as in the 2018 study, showing that the proportion reporting hazardous drinking decreased with age. In both studies, the age difference was particularly striking between the two youngest age groups (18–29 years versus 30–39 years). The same similarities applied to the average AUDIT-C score for the whole sample and when broken down by gender: In 2018, the average for the entire sample (*n* = 16,046) was an AUDIT-C score of 3.3 (SD: 2.1; CI 95%: 3.3–3.3). The average for women was 2.8 (*n* = 8559; SD: 2.1; CI 95% 2.8–2.9) and the average for men was 3.8 (*n* = 7487; SD: 2.1; CI 95% 3.7–3.8).

We collected the data for this study during the second half of April, just after Easter, which tends to be an occasion with increased alcohol use [34]. Additionally, the measures against the COVID-19 pandemic had been put into practice just five to seven weeks earlier, hence the situation was still new and uncertain for those in the sample. Thus, the time of data collection may have influenced the reporting.

Furthermore, data were collected over two weeks. A longer period could ensure less monthly variation. This should be remembered when comparing the findings with other data. Self-reported binge drinking is also commonly subject to monthly variations; female binge drinking appears to be more situation-specific, whereas male drinking is more habitual [35]. As the data collection coincided with Easter, this may have especially affected the results of female binge drinking in the present study.

In terms of strengths, the large sample size in the present study provided analyses with high statistical precision and power. Another strength was that the study period concurred with the most extensive lockdown measures in Norway to date, thus offering a unique insight into a phase that exemplifies the impact of a large-scale pandemic and concurrent lockdown on health-related behaviors, such as alcohol consumption. The use of a validated questionnaire was another asset.

The study limitations should however also be noted. This study employed a cross-sectional design, which limits inferences about directionality and causality and as such prevented us from concluding firmly about the effect of COVID-19 and related exposures on outcomes. It also relied exclusively on self-reporting and might have thus been vulnerable to the common method bias [36]. To assess the change in alcohol consumption, a question that has not previously been validated was constructed. The results may further have been impacted by recall bias and social desirability bias. The response rate with systematic response rate differences between different groups was another limitation, although this was partially compensated for by the use of weights. Furthermore, the questionnaire was written in Norwegian and only distributed through digital means, thus to some degree excluding people without access to the Internet and people with limited proficiency in the Norwegian language. A response rate of 36% can be considered reasonable for an online survey. The response rate in online surveys is approximately 11% lower than that of other survey methods [37] and mostly ranges from 20 to 30% [38]. A meta-analysis found that across 39 comparative results, the unweighted average response rate of online surveys was 34% [39]. Considering the high prevalence of hazardous drinkers in this study, the cut-off score of the AUDIT-C should perhaps be reconsidered. Cut-off scores have been discussed previously [40,41]. Rather than concentrating on the total score, it might be useful to look at the responses separately and calculate the weekly alcohol use and frequency of binge drinking [41]. The AUDIT-C assesses self-reported information on alcohol consumption during the past 12 months. However, when completing the questionnaire greater importance is typically given to more recent events; hence, recency bias has to be taken into consideration when interpreting the results. It is also a difficult task to accurately recall alcohol consumption from a whole year ago. Additionally, self-reported alcohol consumption often comes with an inherent limitation due to underreporting [42,43]. People who decline to participate and people who are not invited to surveys are often characterized by more heavy drinking than those who respond [44,45]. The content of the questions in this study may have also contributed to selection bias, as hazardous drinkers may have been reluctant to participate because of concern about being identified. However, this study was not presented as an examination of alcohol habits but rather as a study of experiences in general during the COVID-19 epidemic in April 2020.

## 5. Conclusions

In conclusion, more than half of respondents reported hazardous drinking behavior and one-tenth reported increased alcohol consumption during the pandemic lockdown period. Increased alcohol consumption was particularly common in the age group of 30–39 years, among people with economic worries due to COVID-19, and among those who were placed in quarantine or working or studying from home. This could be important information for policymakers to keep in mind when revising measures to tackle pandemics.

## Figures and Tables

**Figure 1 ijerph-18-01220-f001:**
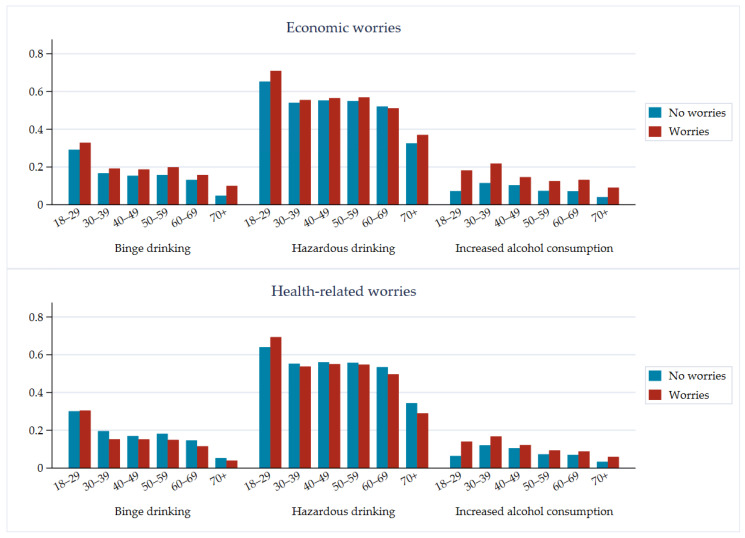
Distribution of binge drinking, hazardous drinking, and increased alcohol consumption in relation to age and worries (inverse probability-weighted estimates). The differences between those worried and those not worried were significant (*p* < 0.05 for each of the three outcomes).

**Table 1 ijerph-18-01220-t001:** Background information about participants per age group.

Age	18–29	30–39	40–49	50–59	60–69	70+	Total *n* (%)
***N***	3347 (13%)	4167 (16%)	4733 (18%)	5292 (21%)	4510 (18%)	3659 (14%)	25,708 (100%)
Gender (women)	2166 (65%)	2501 (60%)	2738 (58%)	2989 (56%)	2318 (51%)	1740 (48%)	14,452 (56%)
Primary school	424 (13%)	157 (4%)	188 (4%)	282 (5%)	399 (9%)	474 (13%)	1924 (8%)
High school	1196 (36%)	787 (19%)	952 (20%)	1647 (31%)	1463 (33%)	1201 (33%)	7246 (28%)
University ≤ 3 years	871 (26%)	1067 (26%)	1128 (24%)	1277 (24%)	1000 (22%)	814 (23%)	6157 (24%)
University > 3 years	844 (25%)	2132 (51%)	2448 (52%)	2064 (39%)	1630 (36%)	1128 (31%)	10,246 (40%)
Adjusted income (EUR) *							
0–25,000	1054 (36%)	537 (13%)	478 (11%)	377 (8%)	251 (7%)	383 (13%)	3080 (13%)
25,000–50,000	1116 (38%)	1977 (50%)	2301 (51%)	1839 (38%)	1380 (36%)	1438 (50%)	10,051 (44%)
>50,000	739 (25%)	1474 (37%)	1700 (38%)	2596 (54%)	2229 (58%)	1051 (37%)	9789 (43%)
Persons in household							
1	477 (14%)	617 (15%)	514 (11%)	880 (17%)	1234 (29%)	1460 (43%)	5182 (21%)
2	1176 (36%)	886 (22%)	615 (13%)	1618 (31%)	2208 (51%)	1554 (46%)	8057 (32%)
3–4	1223 (37%)	1964 (48%)	2314 (50%)	2144 (42%)	800 (18%)	331 (10%)	8776 (35%)
5+	433 (13%)	649 (16%)	1210 (26%)	510 (10%)	83 (2%)	50 (1%)	2935 (12%)
Employment	2206 (66%)	3597 (86%)	4241 (90%)	4650 (88%)	2523 (56%)	230 (6%)	17,447 (68%)
Student/school	1607 (48%)	257 (6%)	105 (2%)	32 (1%)	7 (0%)	3 (0%)	2011 (8%)
Placed in quarantine	747 (22%)	681 (16%)	716 (15%)	744 (14%)	682 (15%)	603 (16%)	4173 (16%)
Temporarily laid-off	513 (15%)	401 (10%)	362 (8%)	419 (8%)	225 (5%)	20 (1%)	1940 (8%)
Home office/study	2217 (66%)	2711 (65%)	3179 (67%)	2909 (55%)	1476 (33%)	154 (4%)	12,646 (49%)
COVID-19 symptoms	279 (8%)	353 (8%)	376 (8%)	328 (6%)	165 (4%)	80 (2%)	1581 (6%)
Worries	2209 (66%)	2499 (60%)	2510 (53%)	2861 (54%)	1857 (41%)	1145 (31%)	13,081 (51%)
Worries related to economy	978 (29%)	1009 (24%)	881 (19%)	866 (16%)	364 (8%)	81 (2%)	4179 (16%)
Health-related worries	1850 (55%)	2051 (49%)	2165 (46%)	2518 (48%)	1684 (37%)	1099 (30%)	11,367 (44%)

* The adjusted income is the household income divided by the personal index. The personal index is calculated as 1 for the first adult, 0.7 per other adult household member, and 0.5 per child. The adjusted income was converted to Euros.

**Table 2 ijerph-18-01220-t002:** Alcohol consumption, binge drinking, and increase in alcohol consumption in relation to age for all respondents, as well as for women and men separately.

Age	18–29	30–39	40–49	50–59	60–69	70+	Total
Mean units per week (both genders) (SD) **	3.3 (4.8)	2.7 (4.7)	2.9 (4.7)	3.1 (4.5)	3.7 (4.9)	3.2 (5.0)	3.2 (4.8)
Women	2.7 (3.9)	2.1 (4.1)	2.1 (3.2)	2.4 (3.7)	2.7 (3.7)	2.2 (3.8)	2.4 (3.8)
Men	4.3 (6.4)	3.9 (6.7)	3.9 (6.4)	4.0 (5.7)	4.3 (5.6)	3.3 (5.3)	4.0 (6.1)
Binge drinking (both genders) **	973 (29%)	625 (15%)	632 (13%)	721 (13%)	553 (12%)	180 (5%)	3684 (14%)
Women	530 (25%)	233 (9%)	198 (7%)	215 (7%)	125 (5%)	34 (2%)	1335 (9%)
Men	443 (38%)	392 (24%)	434 (22%)	506 (22%)	428 (20%)	146 (8%)	2349 (21%)
Increased alcohol consumption (both genders)	482 (16%)	754 (20%)	706 (16%)	559 (12%)	413 (10%)	191 (6%)	3105 (13%)
Women	298 (15%)	437 (20%)	396 (16%)	326 (12%)	196 (9%)	100 (7%)	1753 (14%)
Men	184 (17%)	317 (21%)	310 (17%)	233 (11%)	217 (11%)	91 (5%)	1352 (13%)
Hazardous drinking (both genders) **	2274 (68%)	2250 (54%)	2632 (56%)	2855 (54%)	2371 (53)	1310 (37%)	13,692 (54%)
Women	1503 (70%)	1344 (54%)	1515 (56%)	1603 (54%)	1191 (52%)	648 (38%)	7804 (54%)
Men	771 (65%)	906 (55%)	1117 (56%)	1252 (55%)	1180 (54%)	662 (35%)	5888 (53%)

** Population-weighted estimates (age, gender, education) for percentages and means. ** Mean units per week were calculated via units per drinking day and drinking days per month. ** Binge drinking = drinking more than six units on one occasion at least once a month. ** Hazardous drinking was defined with an AUDIT-C score > 3 for women and > 4 for men. Tables with cut-offs of 4/5 and 5/6 are also available in the Appendix A.

**Table 3 ijerph-18-01220-t003:** Risk factors for increase in alcohol consumption during COVID-19 pandemic phase assessed with logistic regression utilizing odds ratios and 95% confidence intervals. For the adjusted model, all presented variables were included in the model.

Variables	Unadjusted OR (95% CI)	Adjusted OR (95% CI)
18–29	3.2 (2.6−3.9) *	2.1 (1.7−2.7) *
30–39	4.4 (3.6−5.4) *	3.1 (2.4−3.8) *
40–49	3.2 (2.6−3.9) *	2.3 (1.8−2.9) *
50–59	2.2 (1.7−2.7) *	1.6 (1.3−2.1) *
60–69	1.9 (1.5−2.4) *	1.7 (1.3−2.1) *
70+	1.0	1.0
Female	1.0	1.0
Male	1.1 (0.97−1.2)	1.1 (1.0−1.2)
Temporarily laid-off	1.7 (1.5−2.0) *	1.2 (1.0−1.4) *
Quarantine	1.2 (1.1−1.4) *	1.2 (1.1−1.4) *
Home office/study	1.7 (1.5−1.9) *	1.4 (1.3−1.6) *
Economic worries	1.9 (1.7−2.1) *	1.6 (1.4−1.8) *
Health worries	1.2 (1.1−1.3) *	1.1 (1.0−1.2)

Note. OR = odds ratio, CI = confidence interval. * Significantly different from reference group (*p* < 0.05).

## Data Availability

The data presented in this study are available on request from the corresponding author. The data are not publicly available due to privacy restrictions.

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
