# Peer review of "Alcohol Consumption during a Pandemic Lockdown Period and Change in Alcohol Consumption Related to Worries and Pandemic Measures"

_ijerph, 2021, doi:10.3390/ijerph18031220_

Round 1

Reviewer 1 Report

I enjoyed reading this paper.  I have two minor amendments I would like to see which I believe would enhance the manuscript.

1.  Cut-off points for at-risk drinking.  You have chosen a very conservative cut-off for at-risk drinking especially given the circumstances of the survey.  It would also be informative to present two other cut-offs:   4/5 (females/males) 5/6 (females/males).

2.  The current response rate is 36% for an online survey.  You have acknowledged the inherent skewed nature of survey responses.  It would be informative to put this 36% into context of other online surveys.  Is this a reasonable response rate or not.  

Reviewer 2 Report

This is a timely and important study. It's conclusions are no surprise but it's good to have some actual numbers and a reliable study. One of the great strengths here is the size of the sample. It's hard to argue with almost 30K people. 

For me, the critical weakness is that we lack measures of problem drinking behaviors before the pandemic. We have the self-reports of increased drinking and I agree with the authors that those might tend to understate the full extent of the problem. However, there must be some baseline study of alcohol in Norway that could, in a brief paragraph, be presented for comparison. That would really help put the information presented in context.

However, this is a nice bit of work and is nicely presented.

Reviewer 3 Report

The study aimed to evaluate age-by-sex differences in alcohol consumption in (n= 25,708, 56-57% female 40-69 y old, with high education level and purchasing power). At the time of the survey (electronic online survey questionnaires including AUDIT-C)), one out of two participants showed concerns (worries) about their health conditions and their economy to a lesser extent. Multivariable binary logistic regression (OR, CI 95%) was used to estimate self-assessed increased alcohol consumption and associated variables. Binge/hazardous drinking was particularly important among participants aged 18-29 y (man 38%, women 25%) but an increased alcohol consumption due to lockdown was seen from 18-49 y. The authors concluded that health > economic worries were important determinants of a transient higher (hazardous) alcohol consumption.

Major comments. Argument 1: From Table 2, the highest alcohol consumption seems to be within the 18-29 y group and so, advanced age seems to be a protective factor, yet without proving causality. Argument 2: Furthermore, considering the number of subjects with increased consumption (Table 2) + decreased (Table S1) for the same age segment, the percentage of subjects who did not modify their consumption within each age segment was 57.9 (18-29 y), 40.3, 34.0, 27.6, 23.1, 18.0 (70+), 32.6 (Total), Argument 3:  Increased-to-reduced alcohol consumption ratios were 0.33 (18-29 y), 0.81, 0.78, 0.62, 0.65, 0.41 (70+) and 0.59 (total). These arguments point to the fact that the +70 group is the most suitable control group for binary comparisons (Table 3) since this group is more prone to modify its drinking patterns, even though their health-related worries could be more than their economic ones (See “Figure to authors”, PDF file). Also, please consider an additional logistic regression analysis comparing modifiers (increased) and modifiers (reduction) vs. non-modifiers of alcohol consumption.

Title. Suggestion: If the authors decide to make changes to logistic regression analysis (OR-values, Table 3) as suggested above (+70 y control group), the following title might be more appropriate.: “Sudden changes in alcohol consumption among Norwegians during COVID-19 lockdown”: A prospective electronic survey".

Abstract. Once considering re-analyzing vs. 70+ group (control group), data should be reported in a more quantitative way (include p-values) without sacrificing relevant results. Line 26 replace n= 29,535 for n= 25,708.  

Introduction. The authors should highlight the usefulness of conducting epidemiological surveys during sanitary emergencies and currents studies conducted around the globe related to the same theme (e.g. doi: 10.1177/1039856220943024, 10.1016/j.alcohol.2020.07.006, 10.1016/S2468-1253(20)30159-X, 10.1002/hep.31307, 10.3390/ijerph17134677, 10.1016/j.jad.2020.09.012, 10.1001/jamanetworkopen.2020.22942).

Materials and methods. Please provide the institutional ethics registration code.

  • Figures. Figure 1 should be clearer, although I consider that the information it provides is not very relevant in the way it is presented. In case it is decided to leave, it is suggested to put an asterisk above the "worries" bar (red bar) to distinguish the statistical significance with its "no worries" counterpart (blue bar)
  • Tables. All tables should include superscript letters to highlight statistical differences among age-groups considered +70y group as a control. Please include as footnote factor considered in the adjusted model.

Other aspects

  • Do not forget to describe the meaning of abbreviations the first time they are mentioned.

Round 2

Reviewer 3 Report

Thank you for accepting all my comments, the manuscript improved a lot, congratulations